# DAFT: Data-Aware Fine-Tuning of Foundation Models for Efficient and Effective Medical Image Segmentation

Alexander Pfefferle[1][0009−0003−5457−7526], Lennart Purucker[1][0009−0001−1181−0549], and Frank Hutter[2,1][0000−0002−2037−3694]

[1] University of Freiburg, Freiburg, Germany
[2] ELLIS Institute Tübingen, Tübingen, Germany
`{pfeffera, purucker, fh}@cs.uni-freiburg.de`

**Abstract.** Efficient and effective medical image segmentation supports faster and better decision-making of medical experts. In this work, we propose data-aware fine-tuning (DAFT), a method for enabling efficient and effective inference with foundation models, and apply it to medical image segmentation tasks. Following concepts from meta-learning for algorithm selection and dynamic selection, DAFT aims to fine-tune several versions of a foundation model on subsets of all available data instead of fine-tuning just one larger model. Then, at inference time, we select which fine-tuned model to use for the prediction depending on the distribution of the input data. DAFT enables us to create more efficient and effective models for each subset than when creating one model for all data. In our implementation of DAFT for the "Segment Anything In Medical Images On Laptop" competition as part of the CVPR24 Workshop on "Foundation Models for Medical Vision", we use the EfficientViT architecture, knowledge distillation, and OpenVINO runtime to further improve the inference. Additionally, we optimized the efficiency of our method through a flood of improvements, including an optimized inference runtime, caching, optimizing the docker deployment container, and better inference code. DAFT improved the average dice similarity coefficient from 78.64% to 83.29% and the normalized surface distance from 80.58% to 85.59% compared to the baseline on the test data. Our final submission secured first place on the post-challenge leaderboard. Finally, and more importantly, we improved the average inference speed over the baseline by a factor of 6.5 (14.69 to 2.25 seconds) on the test set.

**Keywords:** Data Aware · Fine Tuning · Efficient · Image Segmentation

## 1 Introduction

Medical experts in various medical applications have to spot and detect patterns in medical images from computer tomography (CT), Microscopy, and X-ray on a daily basis. Clinical applications that rely on image segmentation to detect regions of interest in medical images enable experts to make faster and better

decisions. Such clinical applications can be powered by state-of-the-art image segmentation foundation models like SAM [16] or MedSAM [20].

The problem with foundation models for image segmentation is that they often are large, expensive models, e.g., MedSAM has more than 93 Million parameters and requires more than 10GB RAM when run on CPU. Furthermore, trends like the ever-increasing size of foundation models, as seen in the field of large language models[3], will likely make new image segmentation models only more expensive to use in real-world inference in a clinical application. Yet, critically, medical images are always sensitive patient data. Such images are not easily shared with others and often cannot leave the hospital's network or *even leave an expert's laptop*.

Therefore, it is crucial for the viability and usability of clinical applications to enable image segmentation models that are resource-efficient and effective in supporting the decisions of experts. Our goal is to enable even the most resource-constrained experts to benefit from image segmentation models.

Our goal perfectly aligns with the challenge `Segment Anything In Medical Images On Laptop`, organized by Jun Ma, Yuyin Zhou, Bo Wang, Feifei Li, and Sumin Kim as a part of the CVPR24 Workshop on Foundation Models for Medical Vision. In this manuscript, we, the `automlfreiburg` Team from the University of Freiburg, present *data-aware fine-tuning* (`DAFT`), our proposed method to enable efficient and effective inference with foundation models applied to medical image segmentation tasks to solve the challenge.

`DAFT` aims to fine-tune *several versions* of a foundation model on *subsets* of all available fine-tuning data to produce models that need to understand and remember less while also being more effective for their specific subset's distribution. Then, at inference time, we *select* which fine-tuned model to use for the prediction depending on the distribution of the input data. `DAFT` follows traditional concepts from meta-learning algorithm selection [26,19,3,17,28] and dynamic selection [9,4,6], which we adapted to the age of foundation models.

The rest of this manuscript is structured as follows: the remainder of this section introduces the challenge's background, the approach we used, and related work. In Section 2, we present our method in more detail, describing our fine-tuning pipeline and how we improved the runtime speed of our approach. Section 3 contains the implementation details and our protocol for evaluating submissions. Our results are demonstrated in section 4. Finally, we present our improvements for the post-challenge "performance booster" in section5 before concluding our manuscript.

## 1.1   Competition Background

The `Segment Anything In Medical Images On Laptop` competition challenges participants to create a universal promptable medical image segmentation predictor, that is deployable on a laptop. Hereby, deployable on a laptop means

---

[3] https://ai.meta.com/blog/meta-llama-3/

that we *do not* have access to a GPU and only 8GB of RAM and a CPU with 6 cores.

The desired universal promptable medical image segmentation predictor must be able to produce predictions for a wide variety of medical imaging modalities, including 3D modalities, such as Computer Tomography (CT), Magnetic Resonance Tomography (MR), Positron Emission Tomography (PET), 2D greyscale images like Ultrasonic (US), X-Ray, Optical Coherence Tomography (OCT), Mammography and 2D RGB images like Dermoscopy, Endoscopy, Fundus and Microscopy. The prompts are boxes (2D or 3D) surrounding the area of the to-be-segmented area of the image.

The universal predictor, deployed in a Docker [22] container, is evaluated based on the average of the rank of three metrics: the Dice Similarity Coefficient (DSC), Normalized Surface Distance (NSD), and running time.

The organizers provided a preprocessed dataset we could use for training. Furthermore, they shared a list of additional datasets and a list of pretrained models that we were allowed to use. Both lists were extended and curated by the community up until one month before the submission deadline. Moreover, the challenge was hosted on Codabench [31] with a validation leaderboard with up to 6 submissions per day. The organizers also supported up to six docker submissions on the validation data in total.

When submitting to Codabench, participants would upload the predictions of their model and receive the average DSC and NSD for each modality. The organizers would execute Docker submissions in the evaluation environment, and participants received the predictions and runtime for each data point as well as any error messages.

## 1.2   Our Approach

We implemented `DAFT` for this challenge by following a training protocol of 1) knowledge distillation, 2) general fine-tuning, and 3) data-aware fine-tuning for 11 subsets of the data.

In detail, we defined 11 subsets by separating the data based on the origin of the image, like CT or MR. Then for each subset, we 1) created an EfficientViT [5] backbone for our foundation model by knowledge-distilling and using pre-trained weights; 2) fine-tuned the model on all available data; and 3) fine-tuned only on the training data of the respective subset. Then, at inference, we associated the input image with one of our 11 subsets and selected the respective fine-tuned foundation model for segmenting the input image.

Besides `DAFT`, we implemented a flood of improvements for inference efficiency: using EfficientViT as a faster neural network architecture, an optimized inference runtime based on OpenVINO, caching, optimizing the docker deployment container, and enhancing the inference code.

On the test data, we show that `DAFT` improved the average across all modalities for the dice similarity coefficient from 78.64% to 83.29% and for the normalized surface distance from 80.58% to 85.59% compared to the baseline. Our

performance booster submission secured first place on the post-challenge leaderboard. Finally, and more importantly, we improved the average inference speed over the baseline by a factor of 6.5 (14.69 to 2.25 seconds) on the test set.

### 1.3   Related Work

In general, fine-tuning [29,27,13,8] has become more important in recent years due to the prevalence of large and expensive foundation models that need to be adjusted for specific applications at hand. Specifically, fine-tuning has shown to be extremely powerful for medical image segmentation tasks. MedSAM [20] is a segmentation foundation model for medical images created by the organizers of the competition. It was created by fine-tuning the segment anything model (SAM) [16] on over 1 Million medical images. The creators of MedSAM also released LiteMedSAM[4], a lightweight version of MedSAM that was used as a baseline in the competition.

At the same time, there has been research into making segmentation foundation models faster. One area of research in this regard focuses on finding more efficient architectures, e.g. EfficientViT-SAM [33] or MobileSAM [32] for SAM. Speeding up inference of a model can also be achieved by using a runtime that is better optimized for deployment on certain hardware, e.g. OpenVINO [10,1,34] or the ONNXRuntime [7].

Besides fine-tuning, knowledge distillation [11] enables a model that is being trained to leverage knowledge gained by other models that have been trained before. LiteMedSAM was created by distilling the vision transformer in MedSAM to a TinyViT [30] and performing additional fine-tuning afterward.

Furthermore, DAFT is highly related to meta-learning for algorithm selection [26,19,3,17,28] and dynamic selection [9,4,6]. In the former, a meta-model is learned to select one algorithm from a fixed set of potential algorithms to solve a problem. For example, a specific SAT solver is selected to solve a specific SAT instance. This motivated our approach in that we treat different subsets of the data as different problems that certain foundation models might solve better than others. In dynamic selection, a meta-model selects which model is used to obtain predictions *per data point* of a machine learning task. This specifically motivated our inference setup. So far, to the best of our knowledge, no one applied the concepts of dynamic selection or meta-learning algorithm selection to fine-tuning.

A mixture of experts (MoE) model [23,21] is the closest related work for DAFT with foundation models [15]. But MoE models differ fundamentally as the selection, i.e., routing, happens during the inference and training but not before, as we propose with DAFT.

---

[4] https://github.com/bowang-lab/MedSAM/tree/0c044e9b4a6da58775cb4eb4b483aba3f2df5a45

## 2    Method: Data-Aware Fine-Tuning

The concept of data-aware fine-tuning (DAFT) is to select differently fine-tuned foundation models for different *foundation modalities*.

Foundation modalities are subsets of the data that differ in their distribution for the application of a foundation model. Generally, foundation modalities can be understood as different clusters of data points in the data for which a foundation model would be used. In other words, a collection of data points that are sufficiently similar based on their intrinsic characteristics. For this challenge and the application of medical images, we chose the origins of medical images, e.g., CT or MR, as our foundation modalities.

To then decide which fine-tuned foundation model to select given a new input image, we created a *meta-model*. The meta-model predicts which foundation modality the input image belongs to, which in turn decides which fine-tuned foundation model we select to segment the image. Due to our choice of foundation modalities, the meta-model was extremely simple in this challenge, as detailed in the following subsection. Figure 1 provides a general overview of our method.

### 2.1    Data Subset Selection for Data-Aware Fine-Tuning

For this challenge and the application of medical images, we chose the origins of medical images, e.g., CT or MR, as our 11 foundation modalities. This choice is based on our hypothesis that fine-tuning a foundation model that, for example, focuses only on learning Dermoscopy data might perform better on Dermoscopy data than a model that was trained on both CT and Dermoscopy data. Likewise, we hypothesize that the model fine-tuned only on a subset of the data can be made more efficient at inference as it also only requires a subset of the capacity, enabling us, in principle, to use smaller models or prune fine-tuned models more aggressively.

We shortly investigated subdividing X-ray images into X-ray upper extremity, lower extremity, etc. but stopped due to time constraints. Likewise, we considered creating more general foundation modalities by splitting images by {3D modalities, 2D greyscale, 2D RGB}, {3D, 2D}, or {RGB, not RGB}. For the sake of simplicity, we stick with our original choice and leave it to future work to further subdivide these foundation modalities for medical images. Still, we would like to highlight that more general foundation modalities are likely useful to avoid overfitting.

For data modalities with less obvious human-perceivable differences in the data, like with foundation models for tabular data [12] and time series [2], we suggest using unsupervised learning to cluster the data into foundation modalities, or compute meta-features of the data and cluster these meta-features.

Given a set of foundation modalities, we require a foundation model selector to determine when to use which fine-tuned foundation model for an image. To this end, during the initial phase of the challenge, we created a selector for medical images by training a tabular meta-model to predict the foundation modality of an input image. We also considered training an image classifier to predict the

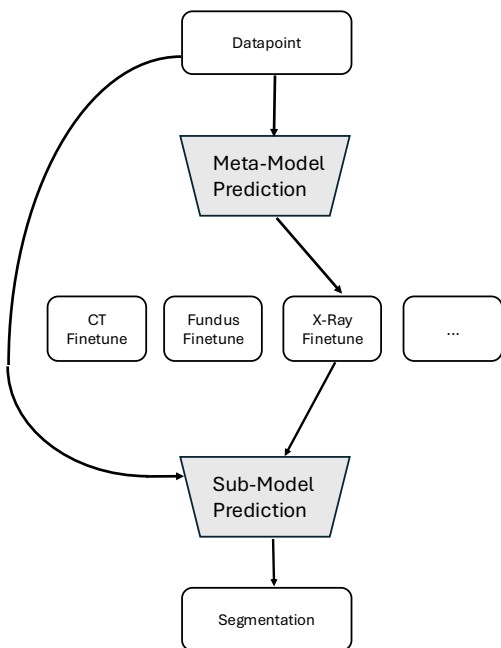

**Fig. 1.** Overview of our implementation of `DAFT` for medical image segmentation.

modality but decided that the increase in runtime would be too expensive. To create a meta-dataset for training the selector, we computed meta-features (like entropy, number of boxes, or percentage of black pixels) of all data points in the training data and stored their foundation modalities (like CT or MR) as target labels. We decided to create two meta-models since the meta-features for 2D and 3D data points can be different. Our model selector checked whether we have a 2D or 3D data point at hand and used the corresponding meta-model afterward. We used a scikit-learn MLPClassifier[5] for both meta-models and achieved an accuracy score of 64% on 2D and 88% on 3D when training on the dataset provided by the organizers and evaluating on the official validation set. This shows that we are able to differentiate the foundation modalities reasonably well with our straightforward meta-features and meta-models.

While the selector was working as intended, we realized that our choice of foundation modalities would always be known in a realistic use case for the medical domain. In real-world settings where medical image segmentation is used, we would know the modality within our software, as the different modalities are clearly separable medical applications (and software products). With this in mind, we instead opted to select the foundation model based on the file name of the image during inference. The file name of all images in the challenge used a naming convention that indicated their origin (e.g., 3D images start with `3DBox_` and 2D images start with `2DBox_` followed by the modality and case number: `3DBox_PET_0001`). We confirmed with the organizers that this approach is allowed and in the spirit of the competition before focusing on it as our final meta-model to predict the foundation modality.

The final implementation of our meta-model is a tree of if-else cases based on parsing the file name and mapping a leaf in the tree to a foundation model fine-tuned on a subset of the data. As the naming conventions were not always consistent and since we believed that there might be unknown naming conventions at test time, we devised several additional naming checks and a general fallback case. The fallback case would use the provided LiteMedSAM baseline model to segment an image.

## 2.2   Fine-Tuning Based on Data-Aware Subsets

To obtain a fine-tuned foundation model for each of the foundation modalities, we set up a fine-tuning pipeline including model distillation [25,11], re-using weights of pre-trained models, general fine-tuning and data-aware fine-tuning.

In detail, our fine-tuning pipeline was a three-step process. The first two steps were done once, and the last step was executed for each of the 11 foundation modalities. The pipeline is visualized in Figure 2 and consisted of the following steps:

---

[5] https://scikit-learn.org/stable/modules/generated/sklearn.neural_network.MLPClassifier.html

1. **Knowledge Distillation:** Distill the TinyViT [30] image encoder of LiteMed-SAM[6] to EfficientViT [5] and copy the weights of the prompt encoder/mask decoder.
2. **General Fine-Tuning:** Fine-tune the initial foundation model from the previous step on the entire dataset of images provided by the organizers. This step makes up for errors or forgetting during knowledge distillation and provides us with a pre-trained-like model.
3. **Data-Aware Fine-Tuning:** Further fine-tune the foundation model from general fine-tuning on a subset of the dataset based on the origin of the image.

We initialized EfficientViT-SAM [33] with its pretrained weights[7] and leveraged the training done for LiteMedSAM by distilling the image encoder of LiteMedSAM to the image encoder of EfficientViT-SAM. Since the architecture of the prompt encoder and mask decoder were the same in both architectures, we were able to copy the corresponding weights after knowledge distillation. We used EfficientViT since we found it to be faster at inference speed than TinyViT. All available data was used for the distillation step. We also considered using MedSAM as a teacher network but decided that the distillation process would take too long.

We added the general fine-tuning, the second step in our pipeline, because we used the EfficientViT-SAM architecture as a backbone. Since the EfficientViT-SAM was not pre-trained on medical images, we first need to guarantee that our distilled model achieves similar general performance on medical images to LiteMedSAM. Thus, we also fine-tuned (or, depending on your perspective, re-trained) the distilled model on the entire dataset. For general and data-aware fine-tuning runs, we froze the prompt encoder and only updated the image encoder and mask decoder. Data-aware fine-tuning directly after distillation would likely perform worse as the foundation model might not be properly adjusted to the general distribution of medical images.

During steps one and two, we only trained on a single random slice of each 3D data point in an epoch. This reduced the training time significantly and also ensured that modalities with deep 3D data points that contained many slices did not dominate the training. In the last step, we trained on all slices of 3D data points if the corresponding subset consisted of 3D data only. Thus, `DAFT` also enabled us to have a more efficient fine-tuning pipeline, especially reducing the time required to obtain production-ready foundation models for the foundation modalities with 2D images or only a small number of data points in their respective subset.

**Pre-Processing, Post-Processing, and Loss Function** For pre-processing, we resized and padded images to a size of $(256, 256)$ and normalized the intensities. In all three steps of our pipeline, we augmented images by flipping

---

[6] https://github.com/bowang-lab/MedSAM/tree/LiteMedSAM

[7] Specifically, we used: https://huggingface.co/han-cai/efficientvit-sam/resolve/main/l0.pt

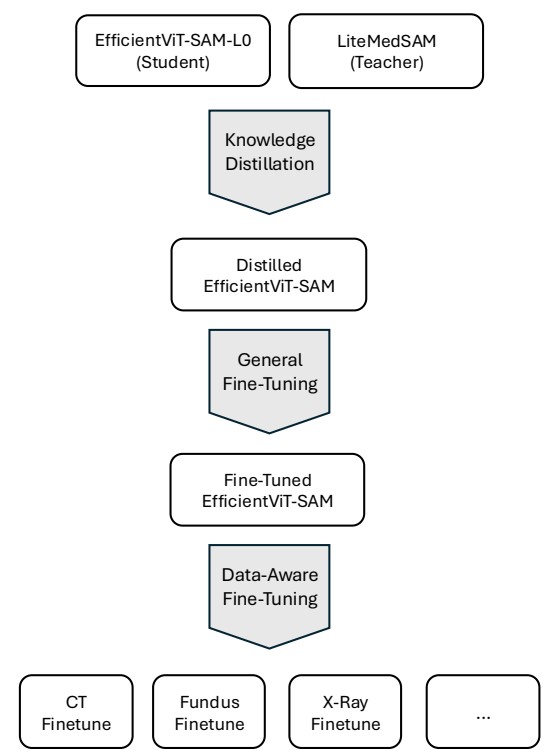

**Fig. 2.** Overview of our fine-tuning pipeline.

them horizontally with a probability of 50% and vertically with a probability of 50%. Following the organizer's baseline, we also randomly increased the size of the prompt's bounding boxes in all directions by up to five pixels. Additionally, for inference, we post-process our prediction by resizing the logits predicted by our model to the original size of the image using bilinear interpolation with a threshold of 0 afterward. Our loss function depended on the step of our pipeline. During knowledge distillation, we minimized the mean squared error between the embeddings predicted by both image encoders. For general fine-tuning and data-aware fine-tuning, we optimized the unweighted sum of the binary cross-entropy loss, dice loss, and intersection over union loss.

### 2.3   Inference Optimization for CPU

We optimized the runtime of our model through a flood of improvements, including using a faster neural network architecture, an optimized inference runtime, caching, optimizing the docker deployment container, and inference code. All our improvements were specifically for CPU or would apply to CPU and GPU.

*Architecture*  We used EfficientViT instead of TinyViT as an image encoder, which made computing image embeddings faster, which is particularly important for 3D images, where we need to compute an image embedding for multiple slices.

*Optimized Inference Runtime*  We replaced PyTorch [24,14] by using the Open-VINO[8] runtime, which made inference faster and also reduced the latency of loading the execution environment before inference. OpenVINO achieves the latter by reducing the number of imports and import dependencies used at inference, avoiding loading the entire PyTorch library, which takes a considerable amount of time. Specifically, this allows us to avoid loading dependencies required only for training. For this challenge, we noticed that reducing loading the execution environment before inference is very important because the docker container will be run once per data point. Hence, if we manage to speed up the latency to the first inference, we save time on every single data point.

*Caching*  Moreover, we used OpenVINO Model Caching[9] to speed up loading our models. This increases the runtime the first time a model is loaded since the cache needs to be created, but all subsequent runs using the same model will be faster since it will be loaded from the system's cache.

*Optimized Docker Container*  We optimized the docker container by reducing its size and the number of layers to increase the efficiency of running commands with the docker container. In detail, we used `python:3.11-slim` instead of `pytorch/pytorch:latest` as a parent image to avoid loading code irrelevant

---

[8] https://github.com/openvinotoolkit/openvino

[9] https://docs.openvino.ai/2024/openvino-workflow/running-inference/optimize-inference/optimizing-latency/model-caching-overview.html

for inference. Using newer Python versions, especially 3.11[10], increases the speed of Python itself. Besides, using a more lightweight version of Python, we opted for the headless version of OpenCV[11] to reduce the number of default packages installed. To further reduce the image size, we specifically removed caching for `apt` and `pip` while building the image. Finally, we combined multiple `RUN` commands and further made sure with docker-squash[12], to get a container that only consisted of a single layer.

The training checkpoints used in the docker image were also converted to more optimized deployment artifacts beforehand. In detail, we converted training checkpoints of all fine-tuned foundation models to ONNX[13] and only then to OpenVINO artifacts.

*Optimized Inference Code* Last but not least, we improved the inference code that was originally provided by the organizers. In this challenge, if given a 3D image, the code first sliced it into multiple 2D images before segmenting each 2D image individually. Then, it would segment each 2D image for each input prompt box *individually* going from the midpoint of the z-dimensions outward in both directions. Thereby, it would use predictions of a prior 2D image as the prompt box for the next 2D image. As a result, the original pipeline would re-compute the image embedding for every 2D image and every prompt box. If two or more provided prompt boxes span across the same sliced 2D images (the same z-dimension of the 3D image), these shared slices would be re-computed for each such prompt box. To optimize this when predicting for 3D images, we avoid redundant computation by caching the image embeddings computed for each sliced 2D image. Thus, we guarantee to compute the image embedding at most once per sliced 2D image across all prompt boxes, i.e., segmentation tasks.

Additionally, we adjusted the original training pipeline for loading 2D or 3D images to work directly on `.npz` files[14]. Without this adjustment, we would need to convert 2D images to `.npy` files and extract and store the sliced 2D images as `.npy` files from the original 3D image `.npz` files.

## 3   Experimental Setup

We follow the experimental design provided by the organizers to obtain results. Additionally, we explain the process behind our development protocol and any remaining implementation details in the following.

---

[10] https://docs.python.org/3/whatsnew/3.11.html#whatsnew311-faster-cpython
[11] https://github.com/opencv/opencv-python
[12] https://github.com/goldmann/docker-squash/tree/fec66e1659e0137d72ea7df57c38a6e36c0fba0b
[13] https://github.com/onnx/onnx
[14] https://numpy.org/devdocs/reference/generated/numpy.lib.format.html

### 3.1   Model Development Evaluation Protocol

For all training runs during development, we restricted ourselves to the dataset prepared by the competition organizers and did not include any other external public datasets. Thus, we also did not include any of the allowed public datasets gathered by the community during the competition's initial phase.

During development, we evaluated the accuracy of our approach by submitting its predictions to the validation leaderboard, treating the leaderboard as our validation data to obtain validation performance. Like in traditional hyperparameter optimization, our evaluation might have overfitted to the validation data as a result of re-using a fixed validation set over the course of the challenge.

To evaluate the runtime during development in a realistic setting, we used the organizer's evaluation script[15] with our docker container on a basic DigitalOcean droplet with 4 vCPUs with 8GB RAM[16]; simulating deployment on a laptop.

### 3.2   Implementation details

Our development code is available on the *finalsubmission* branch of our GitHub repository[17]. The rest of this section details the used environment settings and training protocols, concluding with the results of our training protocol: an overview of the final set of data-aware fine-tuned foundation models.

**Environment Settings**  The training environment and requirements are presented in Table 1. We used this specific environment since it was available on our compute cluster, the JUWELS Hardware Booster[18], which we used for training. Table 2 details the environment we used to create the final model artifacts for deployment in the docker image; we executed this conversion locally on a consumer-grade personal computer. Finally, Table 3 details the requirements used as part of our docker image.

**Training Protocols**  We followed the workflow presented in Figure 1 and described in Section 2.2. Within each step, we optimized for training performance as described in Section 2.2. Finally, we selected the best-fine-tuned model per foundation modality by optimizing for validation performance across all steps of our pipeline. As a result, if general fine-tuning does not improve over knowledge distillation, we stick to the model from knowledge distillation. Likewise, if data-aware fine-tuning does not improve over general fine-tuning, we stick to the model from general fine-tuning.

The details of our training protocols are shown in the following tables: Table 4 presents the protocol for knowledge distillation; Table 5 presents the protocol

---

[15] https://github.com/bowang-lab/MedSAM/blob/0c044e9b4a6da58775cb4eb4b483aba3f2df5a45/CVPR24_time_eval.py
[16] https://www.digitalocean.com/pricing/droplets#basic-droplets
[17] https://github.com/automl/CVPR24-MedSAM-on-Laptop/tree/finalsubmission
[18] https://www.fz-juelich.de/en/ias/jsc/systems/supercomputers/juwels

**Table 1.** Training Environment and Requirements

| System | Rocky Linux release 8.9 (Green Obsidian) |
|---|---|
| CPU | AMD EPYC Rome 7402 CPU, 2× 24 cores, 2,7 GHz |
| RAM | 100GB of 512 GB DDR4, 3200 MHz |
| GPU (number and type) | Four NVIDIA A100 40G |
| CUDA version | 12.0 |
| Programming language | Python 3.11.3 |
| Deep learning framework | torch 2.1.2 |
| Specific dependencies | monai 1.3.2, numpy 1.25.1, opencv-python 4.10.0.84 Branch of efficientvit[A] |

[A]Link to specific branch

**Table 2.** Model Conversion Environment

| Python Version | 3.10.13 |
|---|---|
| Specific dependencies | numpy 1.24.1, openvino 2024.0.0, torch 2.2.0, onnxruntime 1.17.1 efficientvit[A] |

[A]Link to specific branch

**Table 3.** Docker Image Requirements

| Parent image | python:3.11-slim |
|---|---|
| Specific dependencies | numpy 1.26.4, openvino 2024.0.0, opencv-python-headless 4.9.0.80 |

used for general fine-tuning and data-aware fine-tuning (`DAFT`), we used the same protocol and only changed the input data for `DAFT`; and finally Table 6 presents the results of our `DAFT`-based training protocol.

In detail, Table 6 shows how we obtained the final fine-tuned foundation model per foundation modality and the respective number of training epochs. For all but X-ray, Ultrasonic, Dermoscopy, and Endoscopy, `DAFT` improved validation performance. For Endoscopy, not even general fine-tuning improved validation performance in the first place. Likewise, we noticed that if we use only MR or PET data, we start to overfit for MR and PET, respectively. Hence, we used a larger subset of data, merging several foundation modalities for `DAFT` in these two cases. We note that CT, MR, and PET are similar in the images they produce and their application, which motivated merging these specific foundation modalities. Furthermore, we found that none of the models we trained were able to beat LiteMedSAM, the baseline, on ultrasonic data. Thus, we decided to use the LiteMedSAM version provided by the organizers for ultrasonic data instead of our fine-tuned EfficientViT-SAM models.

Furthermore, for X-ray, we trained knowledge distillation and general fine-tuning only on 80% of all data for only 20 epochs and 46 epochs, respectively, due to using an older version of our code for training. We did not use the l0-checkpoint of EfficientViT-SAM during knowledge distillation of Microscopy for the same reason.

**Table 4.** Training Protocol for Knowledge Distillation

| | |
|---|---|
| Pre-trained Model | EfficientViT l0, LiteMedSAM |
| Batch size | 7 |
| Patch size | 256×256×3 |
| Total epochs | 24 |
| Optimizer | AdamW ($\beta = (0.9, 0.999)$, $\epsilon = 10^{-8}$) |
| Initial learning rate (lr) | $5 \cdot 10^{-5}$ |
| Lr decay schedule | ReduceLROnPlateau ($mode = min$, $factor = 0.9$, $patience = 5$, $cooldown = 5$) |
| Training time | 13.7 hours |
| Loss function | mean squared error |
| Number of model parameters | 30M |

## 4 Results and Discussion

We first present the quantitative results in Section 4.1; next, the qualitative results in Section 4.2; followed by the efficiency results in Section 4.3. The quantitative and efficiency results were obtained on the validation set provided by the organizers. Finally we present the results on the final test set in Section 4.4.

**Table 5.** Training Protocol for Fine-Tuning

| Pre-trained Model | Output Model from Knowledge Distillation or General Fine-Tuning |
|---|---|
| Batch size | 96 |
| Patch size | 256×256×3 |
| Total epochs | 24 (in edge cases 20, see Text and Table 6) |
| Optimizer | AdamW ($\beta = (0.9, 0.999)$, $\epsilon = 10^{-8}$) |
| Initial learning rate (lr) | $5 \cdot 10^{-5}$ |
| Lr decay schedule | ReduceLROnPlateau ($mode = min$, $factor = 0.9$, $patience = 5$, $cooldown = 5$) |
| Training time | 7.3 hours for general fine-tuning |
| Loss function | cross-entropy loss + dice loss + inter. over union loss |
| Number of model parameters | 34.8M |

**Table 6.** Number of Epochs per Pipeline Step and Selected Data-Aware Subsets for `DAFT` per Foundation Modality. To avoid overfitting, we combined several modalities for MR and PET. For X-ray, Dermoscopy, and Endoscopy, we did not perform DAFT as it did not increase the validation score. The table does not include the ultrasonic foundation modality because we used the baseline LiteMedSAM model without `DAFT` for ultrasonic data. Note that we were not able to train more epochs for CT, MR, and PET due to time-constrained resources. These modalities are particularly expensive during training.

| Foundation Modality | CT | MR | PET | X-Ray | Dermoscopy | Endoscopy | Fundus | Microscopy | OCT | Mammography |
|---|---|---|---|---|---|---|---|---|---|---|
| Knowledge Distillation | 24 | 24 | 24 | 20 | 24 | 24 | 24 | 20 | 24 | 24 |
| General Fine-Tuning | 24 | 24 | 24 | 46 | 24 | 0 | 24 | 20 | 24 | 24 |
| Data-Aware Fine-Tuning | 4 | 3 | 3 | 0 | 0 | 0 | 24 | 50 | 24 | 24 |
| Used Data-Aware Subsets | CT | CT, MR, PET | CT, MR, PET | - | - | - | Fundus | Microscopy | OCT | Mammography |

### 4.1   Quantitative Results

We present the quantitative results on validation data for the baseline (i.e., LightMedSAM[19]), an ablation study, and our final submission based on `DAFT`.

Our ablation study consists of an EfficientViT-SAM model *without* `DAFT`, that is, we created one general, large foundation model for all foundation modalities by performing knowledge distillation for 24 epochs and general fine-tuning for 24 epochs on the whole dataset. For the ablation study, we used a PyTorch runtime. The ablation study provides insights across our presented results into how well our method would have been without `DAFT`.

Table 7 shows the results on validation data. `DAFT` improved the average dice similarity coefficient from 82.6% to 88.07% and the normalized surface distance from 81.61% to 89.16% compared to the baseline. Our EfficientViT+`DAFT` approach is specifically effective for Microscopy (65.39% to 87.14% NSD) and PET (16.07% to 56.31% NSD). Our proposed method made no improvements for ultrasonic (US) data as we used the baseline model for this data (the observed differences are noise). Yet, the ablation study shows that our EfficientViT model performs much worse for this data, which explains why we failed to improve over the baseline with `DAFT` for EfficientViT. For all other data modalities, we noticed that our ablation study, EfficientViT backbone, improved over the baseline. And `DAFT` further improves over our EfficientViT backbone.

**Table 7.** Quantitative Evaluation Results On Validation Data. The baseline is LiteMedSAM, the ablation study a knowledge-distilled and fine-tuned version of EfficientViT, and our proposed method uses `DAFT` in addition.

| Target | Baseline | | Ablation Study | | Proposed | |
|---|---|---|---|---|---|---|
| | DSC(%) | NSD(%) | DSC(%) | NSD(%) | DSC(%) | NSD (%) |
| CT | 92.19 | 94.77 | 91.09 | 94.58 | 93.14 | 95.48 |
| MR | 89.13 | 92.66 | 86.98 | 91.28 | 88.21 | 91.73 |
| PET | 46.54 | 16.07 | 70.46 | 55.24 | 71.46 | 56.31 |
| US | 94.78 | 96.81 | 83.89 | 88.63 | 94.77 | 96.81 |
| X-Ray | 75.83 | 80.39 | 71.98 | 77.7 | 77.07 | 82.83 |
| Dermoscopy | 92.47 | 93.85 | 94.94 | 96.38 | 94.97 | 96.41 |
| Endoscopy | 96.04 | 98.11 | 95.24 | 97.94 | 96.60 | 98.61 |
| Fundus | 94.8 | 96.41 | 94.75 | 96.4 | 95.59 | 97.16 |
| Microscopy | 61.63 | 65.39 | 78.12 | 84.62 | 80.86 | 87.14 |
| Average | 82.6 | 81.61 | 85.27 | 86.98 | 88.07 | 89.16 |

### 4.2   Qualitative Results

Figure  3 contains examples of good segmentation results on Dermoscopy, Endoscopy, and Fundus data. The corresponding DSC and NSD scores were 97.28%

---

[19] https://github.com/bowang-lab/MedSAM/tree/2a5a0556cabee8a62c8c1ec7e7cd821909adcb0c

and 98.16% for Dermoscopy, 97.83% and 98.29% for Endoscopy, and 97.96% and 98.7% for Fundus. Figure 4 depicts two examples with bad segmentation results. The Mammography example had a DSC of 81.37% and a NSD of 84.58%. The whole 3D CT datapoint had scores of 76.77% and 91.63%. The bad segmentation results show that our predictions are too large and convex, which our model did not seem to expect for these data points.

**Fig. 3.** Examples of Good Segmentation Results: The first row contains a Dermoscopy data point, the second row is an Endoscopy data point, and the last row is a Fundus data point.

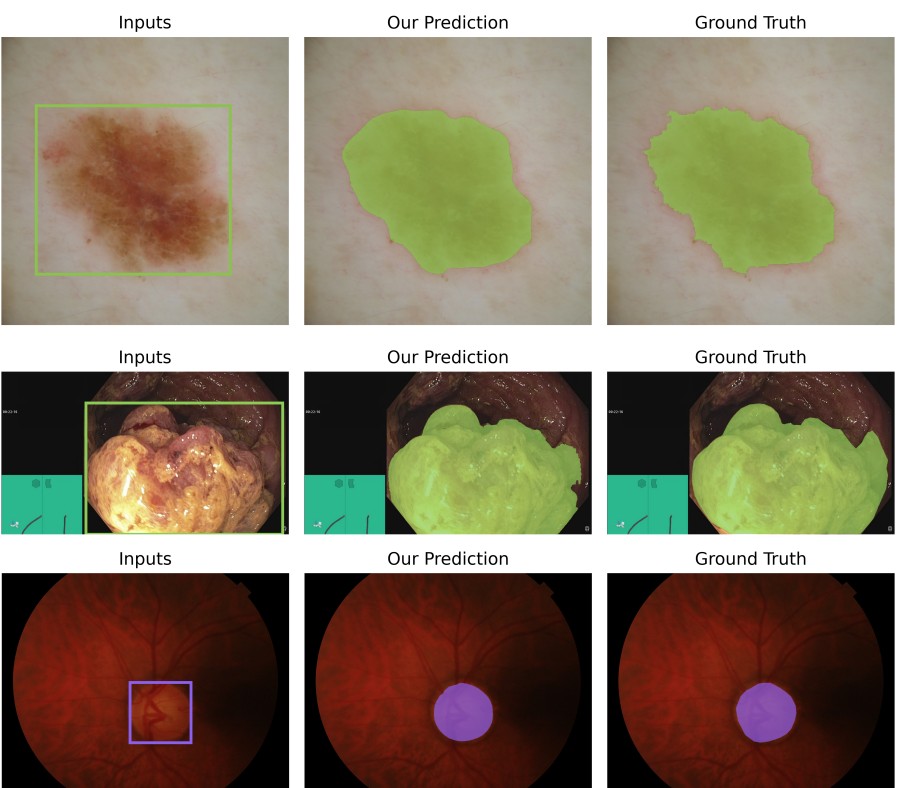

**Fig. 4.** Examples of Bad Segmentation Results: The first row contains a Mammography example, and the second row a slice of a CT example.

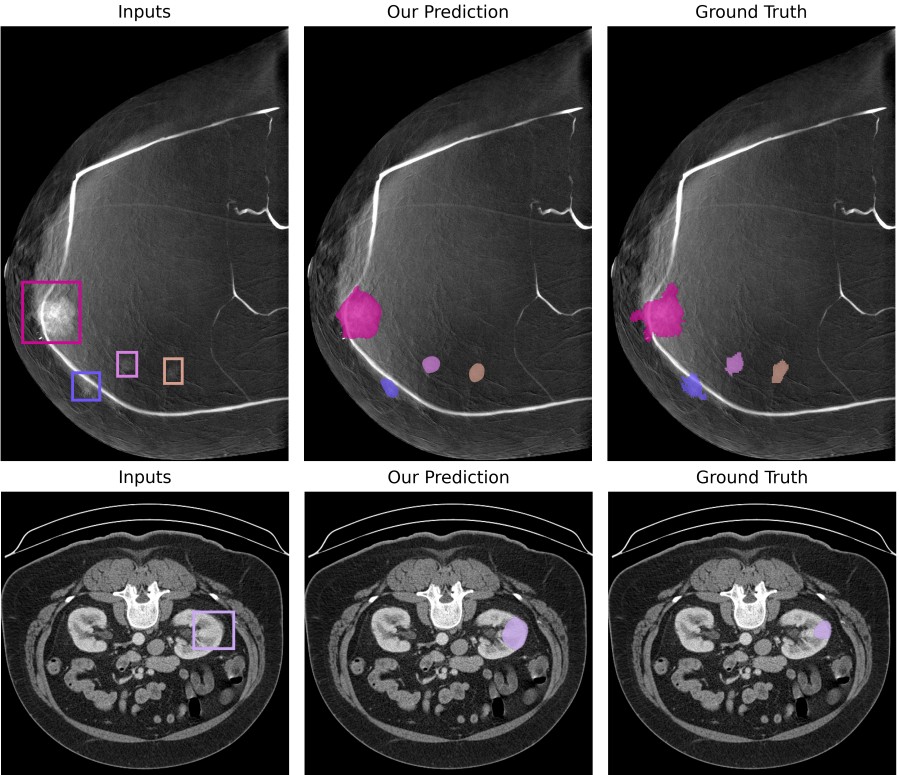

### 4.3   Inference Efficiency Results

Table 8 records the runtime of the baseline and our final submission for a list of example data points from the validation set. We observe the biggest relative improvements for 3D images. This likely follows from caching the computation of the image embeddings and using EfficientViT instead of TinyViT as the backbone, since the image embedding is the most expensive part of the network architecture.

We are also significantly faster on 2D data points. This is likely because our code initializes faster since we replaced the heavy PyTorch library with OpenVINO; this drastically reduced the latency to the first inference. This is more important for 2D data points than for 3D data points, as the inference time of the image encoder is the dominating factor for 3D data points. Our results for ultrasonic (US) 2D data points, where we used the baseline model, also show that our improvements do not only come from using EfficientViT but from the flood of our improvements described in Section 2.3.

For the presented ablation study w.r.t. inference efficiency, we ran our proposed model but replaced OpenVINO with ONNXRuntime[20]. ONNXRuntime is over two times slower on all 3D data points except `3DBox_MR_0121` and also slightly slower on all 2D data points. This shows that OpenVINO with caching dominates the ONNXRuntime for this application.

**Table 8.** Quantitative evaluation of segmentation efficiency in terms of running time (s). We used our own evaluation on CPU, as described in Section 3.1, to obtain the running time per method.

| Case ID | Size | Num. Objects | Baseline[21] | Ablation Study | Proposed |
|---|---|---|---|---|---|
| 3DBox_CT_0566 | (287, 512, 512) | 6 | 814.9 | 266.2 | 113.9 |
| 3DBox_CT_0888 | (237, 512, 512) | 6 | 219.8 | 89.6 | 38.2 |
| 3DBox_CT_0860 | (246, 512, 512) | 1 | 40.2 | 22.6 | 10.2 |
| 3DBox_MR_0621 | (115, 400, 400) | 6 | 389.0 | 96.6 | 45.5 |
| 3DBox_MR_0121 | (64, 290, 320) | 6 | 247.6 | 56.1 | 42.5 |
| 3DBox_MR_0179 | (84, 512, 512) | 1 | 38.3 | 24.2 | 11.1 |
| 3DBox_PET_0001 | (264, 200, 200) | 1 | 30.5 | 16.8 | 7.8 |
| 2DBox_US_0525 | (256, 256, 3) | 1 | 11.3 | 3.7 | 3.5 |
| 2DBox_X-Ray_0053 | (320, 640, 3) | 34 | 13.1 | 5.9 | 5.0 |
| 2DBox_Dermoscopy_0003 | (3024, 4032, 3) | 1 | 10.8 | 4.7 | 3.3 |
| 2DBox_Endoscopy_0086 | (480, 560, 3) | 1 | 11.0 | 4.0 | 2.7 |
| 2DBox_Fundus_0003 | (2048, 2048, 3) | 1 | 11.4 | 3.8 | 3.0 |
| 2DBox_Microscope_0008 | (1536, 2040, 3) | 19 | 13.1 | 5.3 | 4.1 |
| 2DBox_Microscope_0016 | (1920, 2560, 3) | 241 | 35.8 | 29.4 | 27.5 |
| Average Runtime | - | - | 134.8 | 44.9 | 22.7 |

### 4.4 Results on final testing set

Table 9 contains our results on the final testing set. The average DSC across all modalities improved from 76.1% to 79.84% and the NSD from 78.63% to 82.35%. The runtime decreased significantly across all modalities. The average runtime improved from 22.81 to 4.01, which demonstrates a speedup of factor 5.7. When looking at the specific modalities, we observe the biggest increases in accuracy for CT and OCT. We can also observe significant improvements in both DSC and NSD for MR, PET and Microscopy data. We see a significant decrease in accuracy for X-Ray data. Our final submission achieved second place on the testing leaderboard.

---

[20] https://onnxruntime.ai/
[21] We used the code at https://github.com/bowang-lab/MedSAM/tree/2a5a0556cabee8a62c8c1ec7e7cd821909adcb0c and fixed a bug that caused overlays to be saved no matter whether `-save_overlay` was present

**Table 9.** Quantitative Evaluation Results On final testing set. The baseline is LiteMed-SAM, and our proposed method uses `DAFT` in addition.

| Target | Baseline | | | Proposed | | |
|---|---|---|---|---|---|---|
| | DSC(%) | NSD(%) | Runtime | DSC(%) | NSD (%) | Runtime |
| CT | 55.4 | 58.34 | 43.58 | 74.92 | 80.41 | 7.33 |
| MR | 64.83 | 62.84 | 18.75 | 73.5 | 69.22 | 3.58 |
| PET | 61.35 | 57.93 | 84.4 | 66.96 | 60.33 | 10.62 |
| US | 85.25 | 89.73 | 10.72 | 85.29 | 89.73 | 3.72 |
| X-Ray | 85.75 | 94.03 | 9.07 | 71.42 | 81.84 | 2.21 |
| OCT | 67.23 | 73.33 | 7.74 | 79.05 | 85.71 | 2.42 |
| Endoscopy | 94.41 | 96.95 | 6.8 | 94.08 | 96.68 | 1.91 |
| Fundus | 86.33 | 88.39 | 8.05 | 86.74 | 88.79 | 1.98 |
| Microscopy | 84.36 | 86.15 | 16.19 | 86.6 | 88.48 | 2.32 |
| Average | 76.1 | 78.63 | 22.81 | 79.84 | 82.35 | 4.01 |

### 4.5   Limitations and Future work

The biggest limitation of our final submission is the amount of training and validation data we used in our model development protocol. The validation dataset was missing Mammography and OCT data and only had a few data points for certain modalities (e.g., ten for Fundus, or only three for 3D PET); since we used the validation scores to pick our final set of models, we are likely overfitting to the validation data. Likewise, we might overfit to our training data, as our training data was quite limited (e.g., Microscopy had only 1000 data points during training). Furthermore, due to our focus on `DAFT`, we did not perform large-scale re-training or fine-tuning runs across a collection of all publicly shared training datasets, which would likely have resulted in further improvements.

An interesting area for further research is automatically determining the subsets used for data-aware fine-tuning. We initially explored creating a meta-dataset by computing the meta-features of our data points. Afterward, we could compute clusters in the metadata and use the data points corresponding to a cluster as a subset for `DAFT`. Likewise, exploring applications of `DAFT` in application areas, such as tabular data, time series, or NLP, seems very promising.

## 5   Post-Challenge Performance Booster

Following the announcement of the competition results, the organizers invited participants to retrain their models on an enlarged dataset and incorporate potential improvements to try and beat their old submission. They also added data to the validation and test set.

### 5.1   Changes

We incorporated early stopping into all three training steps to avoid overfitting. The provided dataset was split into 80% for training and 20% for validation for

each foundation modality. Then, we used a patience of 7 for early stopping and selected the checkpoint with the lowest validation loss. Additionally, we decided to use a shared model for all 3D modalities, including CT. In section 4.4 we saw that LiteMedSAM ranked bad in accuracy on ultrasonic data in the final test set, so we decided to also use the EfficientViT architecture and DAFT for ultrasonic images. Details about the number of epochs trained are provided in Table 10 and Table 11. Lastly, we wrote the inference code in C++ instead of Python to further improve inference speed. To this end we used the C++ implementation of the winner of the competition, MedficientSAM [18], and modified it to work with our approach. Our updated code is available on the *main* branch of our GitHub repository[22].

**Table 10.** Epochs and $CO_2$eq (g) of Knowledge Distillation and General Fine Tuning. $CO_2$eq is based on 475g $CO_2$/kWh.

| Pipeline Step | Total Epochs | $CO_2$eq | Best Epoch |
|---|---|---|---|
| Knowledge Distillation | 25 | 4050.97 | 17 |
| General Fine-Tuning | 25 | 2020.39 | 17 |

**Table 11.** Epochs and $CO_2$eq (g) of the Data-Aware Fine Tuning step and Selected Data-Aware Subsets for DAFT per Foundation Modality. $CO_2$eq is based on 475g $CO_2$/kWh.

| Foundation Modality | Total Epochs | $CO_2$eq | Best Epoch | Used Data-Aware Subsets |
|---|---|---|---|---|
| 3D | 13 | 26277.3 | 5 | CT, MR, PET |
| X-Ray | 17 | 580.96 | 9 | X-Ray |
| Dermoscopy | 11 | 126.96 | 3 | Dermoscopy |
| Endoscopy | 9 | 394.69 | 1 | Endoscopy |
| Fundus | 13 | 39.38 | 5 | Fundus |
| Microscopy | 25 | 52.71 | 17 | Microscopy |
| OCT | 15 | 28.44 | 7 | OCT |
| Mammography | 10 | 24.44 | 2 | Mammography |
| US | 13 | 30.91 | 5 | US |

## 5.2   Results

We compare the results of our booster submission to the LiteMedSAM baseline in Table 12. We observe that our runtime improved accross all modalities, bringing

---

[22] https://github.com/automl/CVPR24-MedSAM-on-Laptop

the average down from 14.69 to 2.25, which equals a speedup of factor 6.5. We improved the average DSC from 78.64% to 83.29% and the average NSD from 80.58% to 85.59%. The biggest change in accuracy happened for the CT modality, the DSC increased from 55.75% to 73.53% and the NSD from 58.48% to 78.4%. Our performance booster submission achieved first place on the post-challenge leaderboard.

**Table 12.** Quantitative Evaluation Results of our Booster Submission on final testing set.

| Target | Baseline | | | Booster Submission | | |
|---|---|---|---|---|---|---|
| | DSC(%) | NSD(%) | Runtime | DSC(%) | NSD (%) | Runtime |
| CT | 55.75 | 58.48 | 38.78 | 73.53 | 78.4 | 5.59 |
| MR | 64.80 | 62.75 | 18.57 | 72.84 | 70.36 | 2.81 |
| PET | 76.94 | 66.98 | 14.90 | 78.75 | 69.38 | 2.4 |
| US | 85.24 | 89.73 | 8.96 | 89.32 | 93.34 | 1.6 |
| X-Ray | 85.51 | 94.40 | 9.95 | 83.94 | 93.87 | 2.08 |
| OCT | 73.31 | 80.20 | 8.39 | 81.64 | 88.75 | 1.32 |
| Endoscopy | 94.41 | 96.95 | 7.56 | 94.24 | 96.85 | 1.25 |
| Fundus | 87.47 | 89.58 | 8.77 | 86.36 | 88.53 | 1.38 |
| Microscopy | 84.36 | 86.15 | 16.34 | 89 | 90.84 | 1.84 |
| Average | 78.64 | 80.58 | 14.69 | 83.29 | 85.59 | 2.25 |

## 6   Conclusion

In this paper, we proposed data-aware fine-tuning (`DAFT`), a method for enabling efficient and effective inference with foundation models, and apply it to medical image segmentation tasks as part of the "Segment Anything In Medical Images On Laptop" competition. Following concepts from meta-learning for algorithm selection and dynamic selection, `DAFT` aims to fine-tune several versions of a foundation model on subsets of all available data instead of fine-tuning just one larger model. Then, at inference time, we select which fine-tuned model to use for the prediction depending on the distribution of the input data. In our implementation of DAFT we use the EfficientViT architecture, knowledge distillation, and OpenVINO runtime to further improve the efficiency and effectiveness of inference.

In our experiments on the validation data provided by the competition, we show that `DAFT` enables us to create more effective models for each subset than when creating one model for all data. Moreover, we show that we can outperform the baseline by a wide margin. Likewise, we detail the large improvement in inference obtained by our implementation.

Our results show the potential of `DAFT` and optimizing foundation models for inference. Both concepts enable us to deploy efficient and effective segmentation foundation models on the laptops of medical experts.

**Acknowledgments** We thank all the data owners for making the medical images publicly available and CodaLab [31] for hosting the challenge platform. We also thank all authors of prior work for making their weights and code public. The authors gratefully acknowledge funding by the Deutsche Forschungsgemeinschaft (DFG, German Research Foundation) under SFB 1597 (SmallData), grant number 499552394. Furthermore, the authors gratefully acknowledge the Gauss Center for Supercomputing eV (www.gauss-centre.eu) for funding this project by providing computing time through the John von Neumann Institute for Computing (NIC) on the GCS Supercomputer JUWELS at Jülich Supercomputing Center (JSC). Lastly, the authors gratefully acknowledge funding by the Deutsche Forschungsgemeinschaft (DFG, German Research Foundation) under grant number 417962828. We are additionally grateful to the organizers for setting up the challenge and for the great effort and time put into running the competition.

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

**Table 13.** Challenge Submission Checklist Table.

| Requirements | Answer |
| --- | --- |
| A meaningful title | Yes |
| The number of authors ($\leq$6) | 3 |
| Author affiliations and ORCID | Yes |
| Corresponding author email is presented | Yes |
| Validation scores are presented in the abstract | Yes |
| Introduction includes at least three parts: background, related work, and motivation | Yes |
| A pipeline/network figure is provided | Figure 2, 1 |
| Pre-processing | Page 8 |
| Strategies to data augmentation | Page 8 |
| Strategies to improve model inference | Page 10 |
| Post-processing | Page 8 |
| Environment setting table is provided | Table 3, 2, and 1 |
| Training protocol table is provided | Table 4 and 5 |
| Ablation study | Page 16 and 19 |
| Efficiency evaluation results are provided | Table 8 |
| Visualized segmentation example is provided | Figure 3 and 4 |
| Limitation and future work are presented | Yes |
| Reference format is consistent. | Yes |
| Main text $>=$ 8 pages (not include references and appendix) | Yes |