# OpenReview forum: "DAFT: Data-Aware Fine-Tuning of Foundation Models for Efficient and Effective Medical Image Segmentation"
_thecvf.com/CVPR/2024/Workshop/MedSAMonLaptop — CVPR24 MedSAMonLaptop_

### Official Review · Reviewer_s8X2 · 2024-06-12
**This paper explores the concept of data-aware fine-tuning for the MedSAM on a Laptop Challenge, focusing on fine-tuning various models on 11 specific modalities. These modalities are identified by filenames, and the models are based on the Efficient-ViT backbone, distilled using TinyViT as implemented in LiteMedSAM. The authors also apply general fine-tuning to avoid catastrophic forgetting.**

**Rating:** 8
**Confidence:** 4

**Review:**

The paper is exceptionally well-written, with only one minor typo detected. It provides a comprehensive explanation of the entire pipeline, enabling easy replication of the results. The paper is thorough, presenting a strong motivation for data-aware fine-tuning. The authors include enough quantitative and qualitative examples to demonstrate the performance of their methods on the validation data.

I have only one (optional) suggestion, that would contribute to the completeness of the manuscript. In Tables 4 and 5, the number of FLOPs and CO2 equivalents are missing and denoted as "not tracked". However, it is also possible to estimate these values from either a single forward pass (FLOPs) or from running one epoch of training (CO2 eq) and then extrapolate the CO2 for the whole training.

Overall, I recommend the acceptance of this manuscript as it raises no major or minor concerns.

Minor comments:

Typo on page 3: "on all available" should be corrected to "on all available data."

---

### Official Review · Reviewer_WBGE · 2024-06-14
**DAFT: Data-Aware Fine-Tuning of Foundation Models for Efficient and Effective Medical Image Segmentation**

**Rating:** 8
**Confidence:** 4

**Review:**

Paper Summary:
The authors address the Segment Anything in Medical Images On Laptop challenge via their proposed DAFT strategy: A data-aware fine-tuning strategy in which the authors divide the different modalities and leverage knowledge distillation. They then fine-tune the model on all domains and then a second time on the domain of interest, resulting in one model per image domain (except for the 3D domains which they pooled).

Paper Strengths:
- The paper investigates a range of different settings to come up with their approach. The authors ablate their setting against a variant that only uses general fine-tuning and a well-motivated model selection approach.
- The authors report strong results on the validation dataset outperforming the baseline in many imaging domains.
- The paper is written clearly and provides a good amount of details which helps in understanding the approach of the authors.
- The paper uses a "flood of improvements" in which they use various engineering approaches to improve the inference speed of the model.

Paper Weaknesses:
- The motivation as to why the authors use meta-models to select the respective image domain remains a bit unclear. While the now-used approach to only use the image name depends on a substring matching problem which may be dangerous, the authors do not discuss the usage of a simple domain classifier (e.g. ResNet18 with domain classification output)
- The distillation process is not explained thoroughly: which data did you use as input to the teacher model? Furthermore, if you distill, why did you opt to distill from LiteMedSAM over standard MedSAM.

Further Ideas:
- The paper could be further improved by a discussion about the characteristics of the image modalities in which the proposed approach worked really well and in which scenarios the approach failed.

---

### Official Review · Reviewer_qL1N · 2024-06-22
**accept**

**Rating:** 10
**Confidence:** 5

**Review:**

Good paper!!!

---

### Decision · Program_Chairs · 2024-10-01

Accept